# Application of the Catecholaminergic Neuron Electron Transport (CNET) Physical Substrate for Consciousness and Action Selection to Integrated Information Theory

**DOI:** 10.3390/e24010091

**Published:** 2022-01-06

**Authors:** Chris Rourk

**Affiliations:** Independent Researcher, Dallas, TX 75205, USA; crourk@jw.com

**Keywords:** consciousness, catecholaminergic neurons, substantia nigra pars compacta, locus coeruleus, ferritin, electron transport, voluntary action selection

## Abstract

A newly discovered physical mechanism involving incoherent electron tunneling in layers of the protein ferritin that are found in catecholaminergic neurons (catecholaminergic neuron electron transport or CNET) is hypothesized to support communication between neurons. Recent tests further confirm that these ferritin layers can also perform a switching function (in addition to providing an electron tunneling mechanism) that could be associated with action selection in those neurons, consistent with earlier predictions based on CNET. While further testing would be needed to confirm the hypothesis that CNET allows groups of neurons to communicate and act as a switch for selecting one of the neurons in the group to assist in reaching action potential, this paper explains how that hypothesized behavior would be consistent with Integrated Information Theory (IIT), one of a number of consciousness theories (CTs). While the sheer number of CTs suggest that any one of them alone is not sufficient to explain consciousness, this paper demonstrates that CNET can provide a physical substrate and action selection mechanism that is consistent with IIT and which can also be applied to other CTs, such as to conform them into a single explanation of consciousness.

## 1. Application of the Catecholaminergic Neuron Electron Transport (CNET) Physical Substrate for Consciousness and Action Selection to Integrated Information Theory

The binding problem of consciousness can be characterized as the problem of how an integrated, singular conscious experience is created from the large number of neurons and their associated synaptic connections. In this regard, it is important to distinguish the conscious experience from cognitive processing that is often not consciously experienced, but which is necessary for the creation of the conscious experience (such as the visual and auditory cortices, Broca’s area, other sensory cortices and so forth). The integration of cognitive processing from these functionally distinct sections into a single conscious experience is a physical problem, and the nature of consciousness as determined from independent observation is that it relates to a physical substrate that is capable of integrating information into what we experience and observe as a continuous conscious experience.

One of the first references to the “binding problem” per se [1] identified three types of binding: binding by individual neurons, binding by neuronal connections, and dynamic binding. However, approaching binding in this ternary manner is inconsistent with the conscious experience, which is unitary and continuous. For example, to address the dynamic binding problem, a “topographic saliency map that codes for the conspicuousness of locations in the visual field in terms of generalized center surround operations”, that is located in thalamus was proposed, “probably by a winner-take-all mechanism” selected by synchronous oscillations at 40 Hz. However, continuous switching of the experience of consciousness between “winners” would result in discontinuities at least some of the time, which is inconsistent with the continuous nature of the experience of consciousness.

For example, consider a case of seeing a person far away walking towards you on a busy street. At first, the person is too distant to determine whether it is a man, woman or child, but as the person gets closer, you recognize it is a man. Then you recognize it is a young man. Then you recognize it is a young man with glasses. Finally, you recognize it is a young man from a class last year, after you hear he is whistling a tune that he used to whistle in that class. At each of these events (sometimes referred to as a “phase transitions” or “ignitions”), there is continuity in all other aspects of the conscious experience—the traffic noise, perception of the weather, awareness of location, etc. At no point does the entire conscious experience “switch” from one neuron to another, which demonstrates from experience that there is a physical substrate that integrates all of the disparate aspects of the conscious experience into a single “thing”, despite the occurrence of phase transitions. However, the concept of such a physical substrate was not initially recognized, and the implicit assumption was that all physical mechanisms that could be relevant to the problem of consciousness must already be known.

Subsequent work generally continued with this approach [2], while focusing on synchronicity. While not explicitly adopting the “winner takes all” mechanism, von der Malsberg proposed *five* solutions to the binding problem: (1) synaptic plasticity, (2) feature detectors, (3) cell assemblies, (4) associative memory, and (5) visual perception. These proposed solutions are important functional characteristics of a normal, functioning brain, and no doubt they at least contribute to cognitive processing that results in the creation of the conscious experience (although at some level, anything that occurs in the brain contributes to the creation of the conscious experience). However, the increased number of proposed solutions is inconsistent with a unitary experience of consciousness, fails to identify a physical substrate that is capable of integrating neural signals, and also fails to acknowledge that the discovery of such a physical substrate might not yet have occurred. While these functions are necessary because they provide cognitive processing that is integrated into the conscious experience, they are not sufficient to also provide the integrating mechanism.

The winner-take-all (WTA) mechanism was subsequently applied to the global neuronal workspace (GNW) by others (e.g., [3]), which is described as being formed by “distributed neurons with long-distance connections, particularly dense in prefrontal, cingulate, and parietal regions, which are capable of interconnecting multiple specialized processors and can broadcast signals at the brain scale in a spontaneous and sudden manner” [4]. This evolution in the original WTA concept to encompass the GNW appears to have been accepted by Koch [5]. However, the prefrontal, cingulate, and parietal regions encompass areas of the brain that have been removed during surgery while patients were under local anesthesia, and where the patients did not experience a loss of consciousness [6] (“cortical removal even as radical as hemispherectomy does not deprive a patient of consciousness, but rather of certain forms of information, discriminative capacities, or abilities, but not of consciousness itself.”). While computer models of the GNW have been made and compared with subjective reports and objective physiological data [7], analysis of scenarios involving removal of such areas does not appear to have been modelled using those computer models to determine whether they would result in similar outcomes. These WTA mechanisms as they have been applied to the GNW appear to relate to cognitive processes that would be integrated into the conscious experience, but not to a physical substrate or integrating mechanism.

One of the first attempts to address the physical substrate aspect of the binding problem [8] proposed a “conscious mental field” or CMF that was different from “any category of known physical fields, such as electromagnetic, gravitational, etc.”, but which was “not proposed as a view of the metaphysical origin and nature of the mind”. This inconsistency was not explained, nor was it acknowledged that the CMF could be associated with a physical substrate that had not yet been discovered. Subsequent attempts to apply electromagnetic fields to the binding problem were also made [9,10,11,12], but one of the main problems with electromagnetic fields as the physical mechanism is that they do not physically interact below the Schwinger limit [13], which is on the order of 10^18^ V/m or 10^9^ T, well above the levels of electric and magnetic fields in neurons. Instead, these fields add by superposition, meaning that they are separate and distinct below the Schwinger limit. While the electromagnetic field of a neuron can interact with other physical objects such as electrons in other neurons, they do so independently with an effect that is a vector sum of their individual components. To the extent that electromagnetic fields are associated with consciousness, it is by allowing objects such as atoms, molecules and neurons to interact with other objects, subject to Debye screening and other known constraints on electromagnetic fields, but they are incapable of integrating information in a manner consistent with the unitary conscious experience. Thus, while electromagnetic fields are also important functional aspects of the brain and contribute to the creation of the conscious experience, they cannot provide the physical mechanism that integrates neural signals.

Quantum entanglement has also been proposed as a physical substrate that can integrate information from large numbers of neurons, but no technical explanation of how that might occur has been provided. For example, it is stated in [14] that quantum entanglement in microtubules “may be directly related to activities at larger scale levels of neurons and neuronal networks through something of the nature of scale-invariant dynamics”, but no explanation of how such events could integrate neural signals from different functional domains is provided. It appears that this entanglement may be theorized to link the states of the microtubules anywhere throughout the brain (and beyond), such that the issue is not only the linkage, but the lack of linkage of the multiplicity of linked microtubules to the neural spiking level of communication among neurons. There appears to be an absence of a clear explanation of a concrete pathway from the neural sensory inputs to a meaningfully-structured microtubule-level representation. Moreover, since all cells have microtubules, there seems to be no differentiation between entanglement in microtubules in the brain and microtubules in diverse bodily cells, whether body cells equally participate in entanglement-based consciousness, or if an external source of microtubules could be coupled via entanglement.

Consider the experience of recognizing whether a ringing telephone is one on a nearby desk or one on a desk farther away requires integration of both auditory neural signals and visual neural signals. As discussed in [15], the SNc receives neural signals from many areas of the brain, such as the auditory cortex through the cerebellum [16] and the visual cortex [17], as well as from many of the cognitive processing loops, and such neural signals would contribute the selection of an action, such as reaching a hand out to pick up the ringing telephone on the nearby desk, or getting up to walk over the ringing telephone on the desk that is farther away. No similar explanation has yet been provided of how such signals could be integrated by entanglement in microtubules, or for what purpose, as such entanglement in microtubules would not contribute to helping a neuron to reach action potential. This lack of specificity has led to a number of criticisms and questions about the impact of quantum decoherence [18,19,20,21], but in the absence of any viable explanation of how signals from different neurons could be integrated using entanglement, it is difficult to address entanglement as a possible physical substrate. In general, this analysis also applies to other quantal approaches to explaining consciousness, which lack specifics on exactly how such neural signals are integrated and how they cause action potential to be influenced in specific neurons. 

As such, it seems reasonable to conclude that there may be a previously undiscovered physical substrate that could solve the integration mechanism aspect of binding problem, and which would in the process address some of the other problematic aspects of consciousness that also remain unsolved. A recent discovery of electron transport in catecholaminergic neurons (or CNET) could provide a physical mechanism that integrates neural signals [15,22,23,24]. In particular, the CNET mechanism allows electrons from different neurons within discrete groups of catecholaminergic neurons of (1) the *substantia nigra pars compacta* (SNc), (2) the *locus coeruleus* (LC), and possibly (3) the ventral tegmental area (VTA), among others, to interact and share state information from those different neurons in an N-dimensional state function, which could explain certain bounding aspects of the conscious experience [15]. Unlike electromagnetic fields, electrons are not subject to the Schwinger limit, and are capable when they are strongly correlated of forming persistent quasiparticles. Quasiparticles are not fundamental particles, but can represent the quantum interactions of fundamental particles that are not capable of direct observation in a manner that has physical characteristics like an actual distinct particle. Quasiparticles can thus integrate information from a large number of fundamental particles. This paper discusses the application of CNET to consciousness in general, and to the Integrated Information Theory (IIT) of consciousness in particular, as an example of how an integrating mechanism and action selection mechanism can be applied to a CT. It is shown that CNET is fully compatible with the concepts of IIT and that it can be applied to IIT and most likely other CTs.

## 2. An Overview of CNET

CNET is not a CT, but rather is an integration mechanism and action selection mechanism. In this regard, it is noted that both “top-down” CTs that start with the fundamental importance of consciousness and “bottom-up” CTs that start at the level of atomic and molecular interactions each attempt to address qualia, whereas CNET does not do so. While qualia may eventually be understood, to do so will likely require an integration of top-down and bottom-up CTs. CNET stands in the middle of such top-down and bottom-up CTs, and has done what none of them have yet done—provide an integrating mechanism and action selection mechanism that has accurately predicted experimental results.

A brief synopsis of CNET from [15] will first be provided. Neurons are biological switches that turn on by generating an action potential in response to pre-synaptic neural signals. CNET modifies this behavior in groups of specific types of neurons, by facilitating (but not independently causing) the generation of an action potential in one of those neurons in response to post-synaptic neural signals. Neural activity has been observed in vivo in association with the SNc that exhibits this behavior, namely, that SNc post-synaptic activity in the striatum leads rather than follows SNc neural activity [25] (“action specificity of striatal ensembles is predominantly inherited from afferents rather than first emerging through local computations in the [basal ganglia].”). An example of the unusual structure of the large dopamine neurons and associated post-synaptic striatal and cortical neurons that accounts for this behavior can be seen in [26] (e.g., Figure 11). 

CNET results from ferritin structures in catecholaminergic neurons, such as those in the SNc, LC and VTA, which can be found in layers outside of neuromelanin organelles (NMOs) and between neurons in glial cells [27,28]. The physical properties of the arrays of ferritin (which is similar to a P-type doped semiconductor) and neuromelanin (a pi-conjugated polymer structure that is similar to an N-type doped semiconductor) in these structures makes them very similar to mesoscopic semiconductor devices, and are found only in catecholaminergic neurons at a concentration sufficient to support electron transport. Experimental evidence from SNc tissue indicates that CNET is present and that it is able to support electron tunneling even in fixed tissue [23]. As such, previously noted problems resulting from quantum decoherence simply do not apply to tunneling in CNET. 

The CNET mechanism is driven by excitons that are created by dopamine metabolism and homeostasis, which are a byproduct of reactive oxygen species. An exciton is a quasiparticle formed by an electron and associated “hole”, which is an atomic or molecular structure that carries a positive charge. The electron from the exciton can be conducted from a source neuron to an adjacent sink neuron if there is a path, which is provided by ferritin in the neurons and adjacent glial cells, as long as the impedance of the path through the sink neuron to the extracellular fluid is lower than the impedance of the path from the source neuron to the extracellular fluid. The impedance is determined by post-synaptic activity, consistent with principles of electrotonic conduction in neurons. The electron transport mechanism was originally postulated in [22] to possibly be caused by coherent electron minibands, as shown below, but the results of testing from [24] demonstrate that it is more likely the result of incoherent electron tunneling. A Mott insulator can be created in the ferritin structures between neurons, which would prevent electrons from being conducted between CNET neurons unless sufficient post-synaptic activity was present in one of those neurons to change the Mott insulator to a conductor.

The formation of Mott insulators in those tests establishes that incoherent tunneling and strong electron–electron interactions occur in disordered ferritin multilayers, like those found outside of NMOs in catecholaminergic neurons (as shown in [23]). The tests demonstrate that electrons from one neuron can sequentially tunnel to an adjacent neuron under the right conditions, or can be blocked by a Coulomb blockade under other conditions, and also demonstrated that the ferritin and NMO structures in fixed SNc tissue at room temperature are capable of supporting such tunneling.

Electrons in a quantum dot (QD) structure that are engaged in strong electron–electron interactions can form persistent quasiparticles [29], which have previously been discussed with regard to biological systems in association with Correlated Dissipative Ensembles [30]. The study of quasiparticles is revealing that they can be “designed” [31], and a large number of different types of quasiparticles have been discovered to date, including excitons, phonons, magnons and polarons [32]. In this regard, it is noted that the definition of a quasiparticle is often a function of the material in which the quasiparticle is found, such that there could possibly be a specific quasiparticle that is unique to ferritin, i.e., a “ferriton”. Quasiparticles can become entangled [33], although the characteristics of persistent entangled quasiparticles do not appear to have been investigated yet. As such, while the specific physical process by which electrons in ferritin can create a Mott insulator and the specific physical characteristics of the quasiparticles that are formed when that occurs have not yet been studied or quantified, it is possible to do so, and at a relatively low cost given the fact that ferritin is an abundant and easily harvested biostructure and can be tested at room temperature using commonly available equipment. However, at a minimum, the ability of the ferritin and NMO structures in fixed catecholaminergic neurons to conduct electrons by tunneling and for highly similar ferritin structures to form Mott insulators at room temperature and under ambient conditions has been demonstrated.

In entanglement, the states of the entangled particles are physically correlated, such that even if the particles are separated by a substantial spatial distance, the measurement of state of one particle will reveal information about the state of the other entangled particle [34]. For example, two electrons that share an atomic orbital are entangled to the extent that their spins are opposite, meaning that if the spin of one of the entangled electrons is measured, then the spin of the other entangled electron will be known. This measurement is non-trivial, because the electrons would need to be separated from the atomic orbital without any interaction with other particles, and then one electron would need to be measured in a way that does not affect the other electron. In addition, the type of measurement that is performed (e.g., to look for “up” spin or “down” spin) can be determinative of the type of measurement that is made. An unsolved question of quantum mechanics is the mechanism by which the type of conscious observation determines the observed quantity, and how that conscious observation information is transmitted essentially instantaneously. Perhaps for this reason, some have concluded that entanglement is necessary for consciousness, but that is a *non-sequitur* because random selections of measurements result in the same outcome as measurements made in response to conscious decisions. While large numbers of particles can be entangled, that property has only been shown experimentally in exotic materials [35], and no attempt has been made to explain how entangled particles could integrate neural signals, as has been done for CNET.

In contrast to entanglement, strong electron–electron interactions result in strongly correlated electrons that can create quasiparticles and involve interactions between the spin states, charge, orbitals and other electron state variables of large numbers of electrons that can result in unusual bulk properties, such as superconductivity and Mott insulator formation [36]. The Schwinger limit does not apply to quasiparticle formation from strong electron–electron transactions, which have been observed at room temperature and ambient electric fields. Observations of strongly correlated electrons have shown that they can change state (e.g., from conducting to superconducting or from conducting to Mott insulators) in what appears to be a simultaneous manner. Unlike entangled electrons, which appear to transmit state information simultaneously regardless of the separation distance, strongly correlated electrons share state information only within a specific medium. The amount of state information that can be stored in an electron wave function is not known, but the interaction of strongly correlated electrons and the state associated with such interactions can be modeled as a quasiparticle [37].

In CNET, the strong electron–electron interactions are caused by the QD nature of ferritin and neuromelanin. As discussed in [24], ferritin is an iron storage protein that contains a core of ferrihydrite and ferrihydrite precursors, as well as magnetite structures. This structure allows “excitons” to form, which are like atoms in that they have negatively charged electrons that are coupled by Coulombic attraction to positively charged “holes”, that are similar to the nucleus of an atom. The electrons of the excitons exist in their wavelike state, which is a probability wave that identifies the probability of measuring the electron at a particular location and a particular momentum. This wavelike state allows the electrons to “tunnel” between ferritin cores, which is the ability of the electron to move over distances that are much greater than would normally be possible (e.g., on the order of nanometers instead of angstroms). 

The CNET hypothesis as proposed in [22] included an example of one way that a switching function could be provided by CNET. Why a switching function? If the CNET mechanism allows neurons to exchange energy, then it would have to be for a function, and the primary function of a neuron is to reach action potential and fire. Routing of energy to aid in action potential is thus similar to way in which chlorophyll functions to route energy from absorbed sunlight to the reaction center of a chloroplast to allow it to be stored in chemical form [38]. A coherent electron miniband mechanism was identified as one possible mechanism that could provide that switching mechanism, and which also made sense as far as being a possible physical substrate associated with the conscious experience goes, because it would allow the electrons to integrate state information and result in the electrons being coherent or sharing state in a manner that could explain the singular conscious experience. However, the tests reported in [24] were not consistent with miniband formation and localization, because a uniform ferritin array structure was not formed. Instead, the multilayer ferritin structures behaved in a manner consistent with the formation of a Mott insulator, where the path formed by the multilayer ferritin structures resulted in a narrowing path in one direction and a broadening path in the opposite direction. When the electrons tunneled through the narrowing path, the available orbitals in the ferritin filled up and formed a “Coulomb blockade”, which refers to electric resistance caused by Coulombic forces between electrons. The Coulomb blockade switched the multilayer ferritin structures from a conductor to a Mott insulator in that direction of electron movement. While that effect was the result of the specific architecture of the circuit that contained the multilayer ferritin structures, it demonstrated that inside of a group of neurons, changing environmental conditions could potentially cause Mott insulators or conductors to form between neurons, to allow them to selectively share energy under conditions where one neuron is more suitably configured to receive it, as discussed further below.

These experimental observations modify the hypothesized mechanism by which switching may occur in the CNET hypothesis from coherent electron conduction and localization to tunneling but do not change other aspects of the CNET hypothesis. As such, it is first necessary to apply these experimental observations to the CNET hypothesis in order to explain how this tunneling and Mott insulator-based switching mechanism can result in the hypothesized switching function in groups of CNET neurons as well as the experience of consciousness. The CNET mechanism will then be applied to IIT, to explain how a physical mechanism associated with the integration of information from multiple neurons is consistent with or modifies the concepts of IIT. 

## 3. CNET Electron Tunneling and Mott Insulators

The neurons involved in CNET can be viewed as electron sources or sinks, and glial cells between the neurons that contain ferritin can be viewed as ferritin bridges. As shown in Figure 1A, this convention allows electron tunneling between neurons to be understood as a sequence of tunneling events between ferritin cores in glial cells. In Figure 1B, the concept of a Mott insulator is shown, where the two electron sources each try to send electrons over the ferritin bridge (ferritin stores iron as Fe^3+^, which can readily accept one electron to form Fe^2+^). Figure 1C shows the relationship of these structures to the CNET hypothesis presented in [22]. Because the ferritin cores can only hold a limited number of electrons, this results in a Coulomb blockade, where the electric field generated by the electrons from each electron source is insufficient to cause electrons to flow in either direction, which can also be referred to as a Mott insulator.

The electrons in this circuit interact with each other in a manner that is unlike electrons in a metallic conductor, which forms a closely packed lattice structure of positive metal ions and delocalized electrons that can move between the orbitals of the metal ions. Because electrons have the same charge, they generally repulse each other in a metal and do not have strong interactions. In contrast, strong electron–electron interactions can occur in QDs and transition metal oxides such as the iron oxides in the ferritin core, and are more difficult to model. The Hubbard model provides a useful approximation for those interactions [39], but a universally accepted model for such interactions that could provide more information regarding the possible number of states of an electron wave function or quasiparticle has not yet been developed. However, the state of each electron/quasiparticle in a neuron is expected to include information about the state of the neuron, including the electric field, the dendritic and axonic impedances and possibly other state information, because such physical parameters are known to directly affect an electron/quasiparticle. Electrons functionally share state information and behave collectively in a manner that can be modelled as a quasiparticle [40].

Evidence was obtained of this mechanism in SNc neural tissue from conductive force atomic microscopy tests [23], as predicted by [22], and it is also hypothesized to be present in the LC and possibly the VTA and other catecholaminergic neuron groups. The interaction between the electrons as they move between neurons through ferritin cores in the glial cells and also between electrons inside of the neurons can allow those electrons to be strongly correlated through strong electron–electron interactions, form quasiparticles and share the state information from each neuron, resulting in integration of the state information from each neuron in the CNET electrons. The CNET mechanism is hypothesized to be used by the neurons to facilitate the firing of an individual neuron that is optimally configured to fire, as further explained in [15]. However, sharing of state information and collective behavior of electrons in a Mott insulator could provide the physical substrate that is responsible for the generation of the conscious experience. Unlike a CT, the CNET hypothesis is directed to an integration mechanism and action selection mechanism and inherently acknowledges that additional discoveries must be made in order to fully explain consciousness, either in the field of quasiparticle formation from strong electron–electron interactions (known unknowns), or possibly in the way in which electron interactions with their environment affects electron wave function characteristics (unknown unknowns).

In this regard, it is noted that stimulating a neuron with an electric probe would not directly interact with these electrons, and would instead only interact indirectly, by altering the electric field distribution and chemical reactions occurring within the neuron. In order to interact directly, a special probe would be needed that either uses ferritin or other QDs to generate excitons that directly interact with the strongly correlated electrons in strong electron–electron interactions. That probe could be similar to probes used for deep brain stimulation (DBS), and experiments could be performed to determine whether direct interactions in that manner are effective at either detecting conscious state information or altering conscious state information, but it might also be the case that the conscious state is a function of other mechanisms that have not yet been determined (such as chemical or other interactions with the electron wave functions).

At the mesoscopic level, the interaction of these neurons can be modelled as an instantaneous state machine that dynamically changes in a continuous manner as conditions at each neuron change. For example, as shown in Figure 2, individual soma could be in at least two different states: (1) active and conducting electrons using the tunneling mechanism, or (2) dormant and unable to contribute to tunneling. The level of exciton activity and associated tunneling is believed to be associated with the level of dopamine production, because dopamine metabolism and homeostasis produces high energy triplet state electrons in SNc neurons [41]. Glial cells between these neurons would either be able to conduct electrons by tunneling, as shown by the solid arrows, or would not be conducting, as shown by the dashed lines. The direction and level of tunneling would be a function of the activity at the post-synaptic neurons, as that would provide the function associated with allowing groups of neurons to cooperate in a selection mechanism to provide energy to the neuron that is optimally configured for performing an action (i.e., the neuron that has the most associated post-synaptic activity from neural signals associated with sensory and cognitive processing). This is a simplified diagram, and the actual distribution of neurons and glial cells is not in a regular grid as shown, but rather is disordered, with stronger and closer connections between some neurons. The distribution is also three dimensional, but the general concept of a state machine can be seen by the distribution of active and dormant neurons, where state information is exchanged between electrons within the neurons as well as in the glial cells between the neurons.

The function of the state machine is to select a neuron to transfer energy to, to assist with the generation of an action potential at that neuron. In the SNc, this selection results in the firing of a single very large dopamine neuron with ~1 million synapses, which is the equivalent of 1000 normal neurons simultaneously firing (these are a very small percentage of the dopamine neurons in the SNc, most of which are much smaller and are not involved in this mechanism). The post-synaptic connections of these neurons are primarily in the striatum, which receives most of its pre-synaptic connections from other neurons that are associated with action planning and other cognitive processing [25,26]. In this regard, the switch requires both inputs and outputs that are stimulated in order to cause one of the neurons to fire, which is why stimulating the dendrites or somata of the SNc neurons alone does not generally result in an action. In order for action selection to occur, neurons associated with specific muscle groups that are needed to implement the associated action must also be active and providing stimulation to the striatum, as otherwise the activation of the SNc neuron would have little effect. Similar processes in the LC and the VTA are associated with activation of those neurons to stimulate neurons associated with cognitive processing. Consistent with the assumption of the binding mechanism in many CTs, the neural signals provided to the SNc, LC and VTA would need to be synchronized in order for the hypothesized selection/switching mechanism in each of those separate groups of neurons to function.

With this state machine model in mind, it can be seen that the extent and level of excitation of groups of neurons in CNET neurons will determine whether a sufficient activation level exists for selection of a neuron by the mechanism. For example, as shown in Figure 3, a dispersed number of active neurons within a larger number of dormant neurons will not be sufficient to transfer energy to one of those neurons to aid in the generation of an action potential, whereas a concentrated number of active neurons can generate sufficient electrons to aid in the generation of an action potential in one of those active neurons.

The state information that is created and shared by the strongly correlated electrons/quasiparticles would thus create an N-dimensional state, where N is at least the number of participating neurons, but which could be larger depending on the types of information that are encoded in each electron. As discussed in [15], this results in a very large number of possible states, enough to explain the number of states that could be experienced by all of the unique humans throughout their lives. Each conscious state would reflect the stimulation of groups of neurons and the available choices that can be made or actions that can be selected, which would be a function of the synchronized neural signals that are received at the CNET neurons from sensory or cognitive processing by other neurons. For example, the decision to pick up a cup of coffee and to take a sip would be based on cognitive processing of sensory neural signals that identify the location of the cup, memory neural signals that identify the cup as your cup, other memory neural signals that identify the liquid in the cup as coffee, other sensory neural signals that identify that your mouth is available to sip the coffee and not full of food, and so forth. The physical integration of those neural signals by the CNET mechanism creates the instantaneous state associated with the experience of making a decision to drink the coffee, and effectively causes the SNc to select one of the already partially stimulated SNc neurons to reach action potential. Post-synaptic activity of that neuron is associated with striatal neurons that are likewise configured to activate the direct pathway to cause motor neurons associated with picking up the cup of coffee and drinking the coffee to reach action potential [25,26,42]. CNET thus extends all other CTs by explaining exactly how neural signals resulting from cognitive processing can result in what is experienced as voluntary action selection, as most CTs only address cognitive processing and do not even address action selection as a conscious function. Notable exceptions are the Centrencephalic Proposal [6] and Passive Frame Theory [43].

In order to determine whether CNET is used as an action selection mechanism, additional experimental evidence would be necessary. However, such experimental evidence could be obtained in a number of ways, such as:

(1) the large SNc neurons in fixed neural tissue could be identified and tested to see if Mott insulator switching is performed as a function of applied voltages and impedances. This testing would be relatively inexpensive, and could be performed with conventional microprobes and semiconductor test equipment.

(2) QD chromophores that are functionalized to be attracted to ferritin could be used in live neural tissue to detect tunneling. The use of functionalized QD chromophores is well known and relatively inexpensive, and they are routinely used to tag molecules in living cells. In this application, the QDs should emit light if they are coupled to ferritin structures that form part of the CNET mechanism, by virtue of participation in electron tunneling events (such QDs could also detect whether ferritin generates tunneling from electrons donated by antioxidants to neutralize ROS, as well as in magnetogenetics experiments). It may be necessary to use light or applied voltages to generate a sufficient level of exciton activity in live neural tissue, and/or to test the neurons with conventional microprobes and semiconductor test equipment. Quenching with selected anesthetics could also be tested, such as anesthetics that interfere with dopamine or iron metabolism.

(3) *C. elegans* or other simple animals could be observed under live conditions with QD chromophores to see if evidence of light emission in specific neurons correlates to action selection.

(4) transgenic animals could be developed to test the hypothesis, such as transgenic mice that are unable to make ferritin and that require ferritin to be supplied intravenously. The ferritin supply could be interrupted to see whether that resulted in a loss of movement, and then provided to different parts of the animal’s brain to see if doing so resulted in the return of movement.

(5) light sensors could be inserted into a live animal and ferritin-functionalized QD chromophores could be used to determine whether light generation is associated with action selection.

These are only a small set of examples of experiments that could be conducted. Unlike many CTs, though, CNET inherently has the ability to be tested for confirmatory evidence of the action selection mechanism using existing analytical techniques.

If the CNET mechanism is responsible for providing the physical substrate of the human experience of consciousness, it would also imply that many other animals are conscious, as ferritin has been detected in dopamine neurons throughout the evolutionary tree and starting with simple animals like *C. elegans*. Additional testing could be performed to determine whether other animals have SNc, LC and other catecholaminergic neuron groups that contain ferritin structures that can conduct electrons and that are used for action selection and cognitive processing. A semiconductor equivalent to a CNET neuron could be also be constructed and used to determine whether inorganic consciousness can be created, although it is noted that the specific physical structure of ferritin (and possibly neuromelanin or similar compounds) in both non-human animals and semiconductor devices would be completely unlike the CNET structure in human neural tissues and the complex neural structures of the human brain that drive them. If CNET is responsible for the human experience of consciousness, it would be because the human brain has evolved to create and control the sequence of states that results in the human experience of consciousness using the CNET mechanism to control action selection. Indeed, the idea that action selection was a driving force behind brain evolution is well accepted [44,45]. “Transferring” consciousness to an animal or a machine would not be possible, because consciousness would be the result of the neural structures of the brain that control the sequence of states of the CNET neurons, and not the CNET neurons themselves or anything else that could be transferred. 

## 4. CNET Applied to IIT Axioms

CNET is a physical mechanism, and as such, would be a necessary but not sufficient aspect of consciousness if it is shown to exist. In addition to CNET, cognitive processing of neural signals to provide the inputs to the CNET mechanism are needed, similar to those discussed in top-down CTs, but as discussed few CTs address the physical mechanism that would be needed to integrate neural signals to create the singular and continuous experience of consciousness and therefore fail to account for one, or how it might impact the theory. In all likelihood, the combination of CNET and a CT will require the CT to be modified to take into account the physical constraints of the integrating mechanism associated with consciousness. As such, it would make a complex task to explain how the CNET hypothesis applies to all of them [46], and to try and synthesize a CT that is compatible with CNET. 

Compounding the problem of applying CNET to many CTs is the difficulty in stating exactly what it is that a given CT stands for. The different CTs generally represent attempts to analyze and objectively explain a complex phenomenon and may thus provide some value, but the level of complexity to try to apply the CNET hypothesis to all of them would still be a significant undertaking. Instead, a single theory will be analyzed in this paper to demonstrate how CNET can be applied to the analysis of other CTs.

The physical mechanism associated with binding is not directly addressed by IIT, which posits that the binding mechanism itself is the observer of the intrinsic information it holds. It is the state and the structure of the mechanisms that forms the integrated information of the subjective experience [47]. IIT predicts that integrated information is intrinsic to the binding mechanism, which does not address what the physical mechanism is that is responsible for integration. It is noted that this approach does not prove that a physical substrate is not needed, it just assumes that “somehow” integrated information results in consciousness. Furthermore, because IIT does not address the physical substrate, it is primarily a cognitive processing theory as it applies to consciousness, as opposed to a theory that explains how the interaction between the neural signals and the physical substrate result in the conscious experience.

While IIT thus does not rigorously address the physical integration mechanism, it is very thorough in developing axioms, postulates and mechanisms that explain the specific cognitive processing boundaries that are related to consciousness and that define IIT and differentiate it from other CTs [48]. This taxonomy is very helpful in understanding exactly what is proposed by IIT, and for that reason, CNET will be applied to the axioms and postulates of IIT, to demonstrate how the CNET physical mechanism is not only consistent with IIT, but can help to put the axioms and postulates of IIT into context. This analysis is not intended to be critical or supportive of IIT, but rather to show that the structured thought processes that went into developing IIT, which were based at least in part on the human experience of consciousness (because it was developed by what are assumed to be conscious humans and not machines or automatons), can be augmented by the CNET hypothesis. A similar analysis could be developed for other CTs, if a sufficiently definite and articulate statement of exactly what the theory is could be developed, because each of those theories is necessarily based at least in part on the human experience of consciousness. 

### 4.1. Axiom: Existence 

IIT: Consciousness exists—it is an undeniable aspect of reality. Paraphrasing Descartes, “I experience therefore I am”.

CNET: The instantaneous state of the strongly correlated electrons/quasiparticles of the CNET mechanism is the real, physical state that creates the conscious experience as a function of cognitively processed neural signals. 

### 4.2. Axiom: Composition

IIT: Consciousness is compositional (structured): each experience consists of multiple aspects in various combinations. Within the same experience, one can see, for example, left and right, red and blue, a triangle and a square, a red triangle on the left, a blue square on the right, and so on. 

CNET: These phenomenological distinctions result from the physical integration of cognitively processed neural signals that result from sensory data, where the cognitive processing generates neural signals that are used by CNET to isolate a red triangle on the left and a blue square on the right, as well as cognitive data derived from that processed sensory data that classify the red triangle as a red triangle and the blue square as a blue square to CNET [49,50,51]. These neural signals are integrated in the CNET neurons by strongly correlated electrons/quasiparticles in the CNET mechanism. Each electron/quasiparticle in each neuron has state, and when it is strongly correlated with electrons/quasiparticles from other neurons those electrons/quasiparticles share state information. The manner in which state information is stored in an electron/quasiparticle or shared between electrons/quasiparticles is not known, but this shared state information provides the physical substrate for phenomenological distinctions, which is integrated by that substrate into a singular conscious experience. In the same way that strongly correlated electrons exhibit simultaneous state changing behavior under certain conditions, such as switching from a conductor to a Mott insulator, they simultaneously share this neuron state information, which is also determinative of whether the electron will tunnel in the direction of the neuron that has the lowest impedance path to ground as well as through the neurons and glial cells that lie in the electron’s path. The path is constantly changing as a function of neural signals, so the state of each electron/quasiparticle is also constantly changing. Each unique state creates an experience, which can include the state of seeing a red triangle on the left and a blue square on the right. To further understand why that state exists, it would be necessary to understand how information is encoded in an electron/quasiparticle and how electrons/quasiparticles share state information, but the fact that the electrons of the CNET neurons are physically able to encode and share that information has been established by the way electrons have been experimentally observed to behave. In this regard, CNET is different from a CT because it is a physical substrate and action selection mechanism, and not a CT.

### 4.3. Axiom: Information

IIT: Consciousness is informative: each experience differs in its particular way from other possible experiences. Thus, an experience of pure darkness is what it is by differing, in its particular way, from an immense number of other possible experiences. A small subset of these possible experiences includes, for example, all the frames of all possible movies.

CNET: Each experience is derived from cognitive processing that generates neural signals that are integrated into an N-dimensional state in the CNET neurons for the purpose of action selection. For example, part of the brain processes spatial locations and generates neural signals associated with the locations. Cognitive processing also provides inputs that are derived from sensory inputs but which can be distinct from sensory inputs. These neural signals are provided to the LC and VTA for selections related to cognitive processing (such as analysis, planning, memory and so forth), and to the SNc for action selection. The composite conscious experience is one that evolved to result in an action selection that will increase the chances of survival of the observer. The available actions as determined from spatial, cognitive, sensory and other neural signals is the “information” encoded into the conscious experience. The experience of pure darkness is a sensory experience, but cognitive processing is also occurring that processes other neural state information, such as prior state information, location information and so forth. Thus, the state of being in pure darkness does not result in action selection if it is an expected state as part of an amusement park ride, but may result in some other action if it is an unexpected state, particularly one that is accompanied by other sensory information (such as sensory information that indicates that a bag has been placed over the observer’s head).

### 4.4. Axiom: Integration

IIT: Consciousness is integrated: each experience is (strongly) irreducible to non-interdependent components. Thus, experiencing the word ‘‘SONO’’ written in the middle of a blank page is irreducible to an experience of the word ‘‘SO’’ at the right border of a half-page, plus an experience of the word ‘‘NO’’ on the left border of another half page—the experience is whole. Similarly, seeing a red triangle is irreducible to seeing a triangle but no red color, plus a red patch but no triangle.

CNET: The N-dimensional state integration performed by the CNET electrons/quasiparticles is not a difference engine but rather a state machine. If the neurons that are processing SONO in the middle of a blank page are providing signals to the CNET neurons, then that forms part of the conscious experience. Additional neural processing may generate an action selection, such as if a prior state established that the observer should push a button if they see the word SONO and appropriate post-synaptic neural signals required for striatal action selection have been generated by cognitive processing. CNET is based in part on differentials with other sensory inputs, but also on state provided by other neural processes. These different neural signals are based on neural structures that evolved over time to facilitate action selection and to improve the chances of a successful observer. The neural signals can be generated by cognitive processing that causes the conscious experience to be altered, such as to fail to see SONO on a blank page because there is a more important object to observe, such as a tiger in the room, but the neural signals generated when light from the word SONO on a blank page is falling on retinal neurons are still be generated and processed. However, they are ultimately suppressed by the rules of the brain and the action selection mechanism to ensure that the most important visual object—the tiger—remains the focus of observation, and that the best selection of available possible actions related to an evolutionarily successful outcome of the interaction with the tiger is made (e.g., shooting the tiger with the gun in the observer’s hand, closing the door to the room if the observer is outside of the room, etc.). 

### 4.5. Axiom: Exclusion

IIT: Consciousness is exclusive: each experience excludes all others—at any given time there is only one experience having its full content, rather than a superposition of multiple partial experiences; each experience has definite borders—certain things can be experienced and others cannot; each experience has a particular spatial and temporal grain—it flows at a particular speed, and it has a certain resolution such that some distinctions are possible and finer or coarser distinctions are not.

CNET: Experience is a function of state, and the current state is dependent on prior states and controls future states. State is also a function of sensory inputs and cognitive processing. For example, you can look at a bookcase for hours and then suddenly remember that you left a $100 bill in the pages of a book in the bookcase. That sudden memory is a function of the numerous parallel cognitive processes that are occurring even while the sensory input of the bookcase is not changing, and may occur after several moments or several hours, depending on the cognitive neurons that are being innervated by the LC CNET mechanism. Subsequent action planning and execution associated with getting up, walking over to the bookcase, picking up the book with the $100 in it is associated with the SNc (with input from the LC and its efferent axonal signals).

## 5. CNET Applied to IIT Postulates

The IIT postulates are described assumptions that are derived from the IIT axioms, and which are asserted to be about the physical substrates of consciousness as they form the basis of the mathematical framework of IIT. These postulates are existence, composition, information, integration, and exclusion, and as such, the analysis of the applicability of CNET to the postulates bears many similarities to the analysis of the applicability of CNET to the axioms.


**Existence**


IIT: Mechanisms in a state exist. A system is a set of mechanisms.

CNET: The system of the human body includes mechanisms that generate sensory neural signals that are processed by the brain and which are delivered to the CNET neurons, as well as cognitive neural signals that are processed by specific rules. For example, in the visual cortex, these rules result in the identification of lines, colors, movement, objects and so forth. These processed visual neural signals are further processed in parallel by cognitive rules in the cerebral cortex to identify objects in the output neural signals from the visual cortex, to identify logical relationships between the objects (e.g., my coffee cup, someone else’s tiger), the significance of objects (the coffee cup contains coffee that I can drink, the tiger is a viscious predator that can kill me), and other neural signals that are generated by rules that are optimized for increasing the likelihood of survival of the subject. The rules can be hard coded like the rules of the visual cortex or synaptically programmable, like the rules for understanding a spoken language.


**Composition**


IIT: Elementary mechanisms can be combined into higher order ones.

CNET: Subjects can perform complex cognitive processing functions that satisfy multiple rules, such as forming a relationship to procreate and also to assist with breathing, drinking, eating, sleeping and fight or flight. The cognitively processed neural signals of an individual are processed by the CNET mechanism to choose an action (or to choose not to act) that will optimize performance of functions associated with rules, including compositions (e.g., obtaining shelter to improve the chances of breathing, drinking, eating and sleeping in the future, getting a better job results in increased ability to breath clean air, drink clean water, eat safe food, sleep in safety and to procreate). However, this is a cognitive processing function and not a consciousness integration function, and results in a series of states that are directed towards creating an expected series of outcomes. The observed outcomes are compared with the expected outcomes, and can be combined with existing rules that are associated with the creation of future states to form modified or new rules to obtain the future state.


**Information**


IIT: A mechanism can contribute to consciousness only if it specifies “differences that make a difference” within a system. That is, a mechanism in a state generates information only if it constrains the states of a system that can be its possible causes and effects—its cause–effect repertoire. The more selective the possible causes and effects, the higher the cause–effect information specified by the mechanism.

CNET: Cause–effect is controlled by rules directed to behavior that results in an evolutionarily successful organism, and the processor of the brain is designed to optimize the performance of these rules. Sensory inputs function as interventions that are processed by these rules and the state of the individual. Action selection is performed by the CNET neuron groups in response to pre- and post-synaptic inputs to perform functions associated with of these rules, and the experience of consciousness is physically correlated to the CNET mechanism. The cognitive processing rules are designed to compare expected results with actual observed results and the experience specific states such as pain or pleasure and to modify the rules based on the differences between the expected and actual results and those experiences.

Information is the way the system cognitively processes sensory data in accordance with the evolutionarily developed rules for a successful individual and society. Information is not created if it is not associated performance of a rule (e.g., we do not care whether an animal is male or female if we are starving and need to eat it to survive, but we care if we are raising animals for domestication). Some information will not have an associated rule but the system can learn a new rule by the interaction of the CNET components. For example, cognitive processing systems associated with the LC may expect a specific result from a specific action, and the action selection and implementation systems associated with the SNc can cause the action. The result of the action can then be analyzed using cognitive processing from the sensory inputs to determine whether the observed result was the expected result and to analyze what might have caused a different observed result if it was not. For example, a glass may hold a clear liquid, and the expected result may be that it is water. When it is tasted, it is determined to hold vodka. A new rule is made to smell a glass of clear liquid before drinking it to determine if it holds water or vodka. 


**Integration**


IIT: A mechanism can contribute to consciousness only if it specifies a cause–effect repertoire (information) that is irreducible to independent components. Integration/irreducibility Q is assessed by partitioning the mechanism and measuring what difference this makes to its cause–effect repertoire.

CNET: Partitions relate to the ability to effectively process sensory inputs to create information in accordance with the cause–effect rules of the brain. For example, the rules of a properly functioning brain will not stop to determine how many stripes are on a tiger if it enters the room, instead they will create a fight/flight response. Thus, the subject will not be consciously aware of the number of stripes on the tiger even though they might be aware that the tiger has stripes and be able to count them under different circumstances. 


**Exclusion**


IIT: A mechanism can contribute to consciousness at most one cause–effect repertoire, the one having the maximum value of integration/irreducibility QMax. This is its maximally irreducible cause–effect repertoire (MICE). If the MICE exists, the mechanism constitutes a concept.

CNET: This mechanism appears to relate to cognitive processing rules (logic) for resolving complex problems. For example, the concept of a mathematical circle can be used to solve a problem of how to make a wheel or a pizza, depending on the specific action that is being selected (making a wagon or making dinner). 

## 6. Conclusions

CNET provides a hypothesis of a physical mechanism that can integrate neural signals from a number of different neurons and, thus, is different from other physical substrates that are relied on by other CTs (where a physical substrate is even addressed and not just assumed). Those CTs are unable to provide a feasible physical explanation of how integration of neural signals could be accomplished. In addition, CNET is associated with an action selection mechanism that allows the complex, massively parallel-processed neural signals of the brain to be processed for the purpose of selecting an action from a large number of possible actions. In that regard, CNET depends on cognitive sensory preprocessing that has been optimized by evolution to provide continuity of states, memory processing, rule formulation and other functions that would not necessarily allow CNET to be applied to a system that was not specifically configured to use it. However, CNET is applicable to at least one of the articulated theories of consciousness—IIT—and answers questions about how it could be applied to the human brain that are not apparent from IIT in the abstract. Further experimental work is needed to provide evidence that CNET results in action selection, but that experimental work is both feasible and relatively inexpensive.

## Figures and Tables

**Figure 1 entropy-24-00091-f001:**
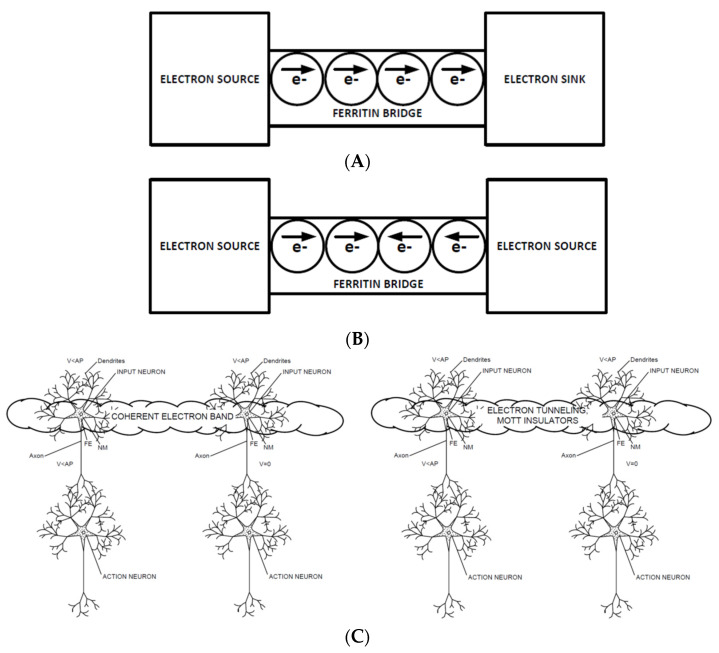
(**A**) electron tunneling between neurons; (**B**) Mott insulator formation between neurons; (**C**) electron tunneling and Mott insulator formation appears more likely following the tests in [24] than the coherent electron minibands that were originally postulated as a potential electron transport and switching mechanism in [22]. The ferritin in the glial cells between SNc neurons provides the physical substrate that allows electrons to tunnel between neurons.

**Figure 2 entropy-24-00091-f002:**
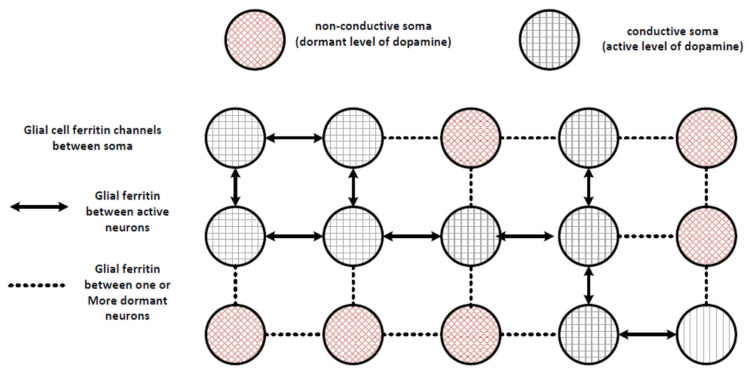
Diagram showing interactions between active and dormant neurons through ferritin channels in glial cells.

**Figure 3 entropy-24-00091-f003:**
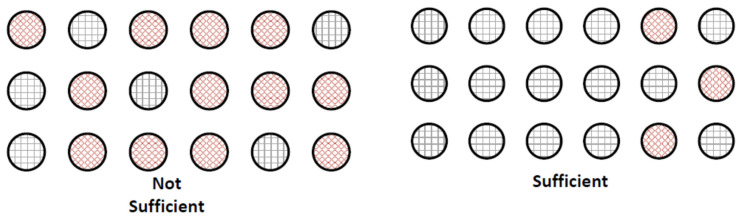
Diagrams showing different simplified neuron states that are not sufficient or sufficient to assist in the generation of an action potential.

## Data Availability

Not applicable.

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
