# Peer review of "Application of the Catecholaminergic Neuron Electron Transport (CNET) Physical Substrate for Consciousness and Action Selection to Integrated Information Theory"

_entropy, 2022, doi:10.3390/e24010091_

Round 1
Reviewer 1 Report
This a well formulated description and proposition of the role and mechanism of a certain type of neurotransmitters, CNET, to integrate signals and communication between a number of different neurons. CNET does focus on electron tunneling in ferritin, found in so-called catecholaminergic neurons supporting communication between them. The mechanism is closely related to the quantum dot generation of excitons, quoting the author, “interacting directly with the strongly correlated electrons in strong electron-electron interactions”. Finally the author makes the case that CNET provides a substrate consistent with and supporting Tononi’s celebrated IIT theory, but leaving conformations open to other consciousness theories, CT, as well.
It is well-known that CT, from the viewpoint of chemical physics, poses some initial problems that are often overlooked. First is the infamous decoherence problem, revealing that open systems in hot and wet environments, such as a human brain, are sensitive to temperature and time scales, not to mention Jacques Monod’s conundrum that communication between life forms goes from objective physical interactions to more subjective correlations that exhibit teleonomic properties.
These are general problems that one does not expect that the present author should be able to solve here and now. However, he refers to previous work in the book series volume Advances in Quantum Chemistry Vol. 82, (2020), with the theme “Quantum Boundaries of Life”. The collection of articles concerns various views of trans-disciplinary explorations of quantum theoretical approaches in life sciences. One would expect due consideration of and a citation to this book, where alternative quantum chemical approaches are also detailed and presented.
The author refers to observations of strong electron-electron interactions changing state from and between superconducting, conducting, and Mott insulators. Although not known in detail, he proposes such interactions to be modeled as a quasiparticle and the possible formation of quantum dots. This is, quoting the author –the wavelike state allows the electrons to “tunnel” between ferritin cores, which is the ability of the electron to move over distances that are much greater than would normally be possible (e.g. on the order of nanometers instead of angstroms) – caused by the quantum dot nature of ferritin. This process appears feasible but it does not follow from the estimates of the thermal de Broglie wavelength. Other mechanisms are needed, see e.g., the ODLCI comment below.
Finally, the author attempts to demonstrate that his model is commensurate with the famed integration-information-theory, IIT, providing conformation to a single explanation of consciousness. Although IIT is very impressing one should realize that it is a top-down hypothesis, starting with the fundamental importance of consciousness, in contrast to bottom-up theories starting at the level of atomic and molecular interactions. Repeating Tononi’s five different postulates, as support for CNET, is too general to be convincing. Bottom-up alternatives, see AQC #82, could yield equally convincing backing, i.e., starting with Yangs concept of ODLRO, C. N. Yang, (1962) “Concept of off-diagonal long-range order, ODLRO, and the quantum phases of liquid helium and of superconductors”. Rev. Mod. Phys. 34: 694-704, via Bloch-Liouville thermalization to the long-range correlation conception of ODLCI.
Conclusion: Although the manuscript contains many interesting ideas and propositions, it needs substantial revision. A detailed repetition of the Tononi postulates is not necessary. Alternative ideas distinguishing top-down and bottom-up models should be separated and a more realistic protocol considering more fundamental issues, based in theoretical chemical physics, should be enunciated, e.g., as recommended in the referee report above.
A technical issue is the need to include appropriate reference numbering.
Author Response
Please see the responses in the attached file.

Reviewer 2 Report
See attached.

Author Response

(The authors gave the same response as above.)

Round 2
Reviewer 1 Report
Even if not all recommendations and suggestions have been accommodated, the revised version is acceptable for publication.
Author Response
The author thanks the reviewer for providing helpful and constructive feedback that has improved the quality and clarity of the paper.
Reviewer 2 Report
See attached

Round 3
Reviewer 2 Report
Here is my review of this latest version of Rourk's text, together with a partially copy-edited version of the text itself.
I have gone through the sections of changed text and made suggested corrections in the .doc file, as this is easier than writing them out verbally.
I could not identify the diagram that the author claims to have added to explain why ferritin in glial cells is needed.
The coding in Fig. 2 is confusing, with dormant cells coded in red, while active cells are shown as white (= empty). I would suggest coding the active cells in red and the dormant cells gray. This coding would then also need to apply to Fig. 3.
I remain unclear what is meant by pre- and post-synaptic. In new text at line 152, the author says: “Neurons are biological switches that turn on by generating an action potential in response to pre-synaptic neural signals. CNET modifies this behavior in groups of specific types of neurons, by facilitating (but not independently causing) the generation of an action potential in one of those neurons in response to post-synaptic neural signals. Neural activity has been observed in vivo in association with the SNc that exhibits this behavior, namely, that SNc post-synaptic activity in the striatum leads rather than follows SNc neural activity (Park et al., 2020) …”
To my understanding, pre-synaptic spikes travel along axons to synapses on the dendrites and cell bodies, causing neurotransmitter release and post-synaptic potential increases, or signals, that lead to the generation of spikes in the axon of the post-synaptic neurons. Where is the site of action of the CNET – at the cell body? How can there be “SNc post-synaptic activity” in a different neural substrate – the striatum? The post-synaptic activity has to be in the cell bodies where they reside.
At line 508, the author claims that CNET is necessary for consciousness, but I think it should be rephrased as “CNET is a physical mechanism, and as such, would be a necessary but not sufficient aspect of consciousness if it could be shown to exist.”
Section 4.1. I don’t see how the CNET text relates to IIT’s first axiom, and the issue is further compounded by the new emphasis on cognitive processing. There is nothing in the CNET mechanism that singles out cognitive processing per se. I suggest deleting the whole paragraph except for the original version of the last sentence, which does seem to relate to the existence of the CNET mechanism (without the cognitive overlay).
Section 5 seems largely redundant with Section 4. I suggest deleting it entirely.

Author Response
See attached file for detailed responses.
